# Compositional Zero-Shot Learning via Fine-Grained Dense Feature Composition

**Dat Huynh**
Northeastern University
huynh.dat@northeastern.edu

**Ehsan Elhamifar**
Northeastern University
eelhami@ccs.neu.edu

## Abstract

We develop a novel generative model for zero-shot learning to recognize fine-grained unseen classes without training samples. Our observation is that generating holistic features of unseen classes fails to capture every attribute needed to distinguish small differences among classes. We propose a feature composition framework that learns to extract attribute-based features from training samples and combines them to construct fine-grained features for unseen classes. Feature composition allows us to not only selectively compose features of unseen classes from only relevant training samples, but also obtain diversity among composed features via changing samples used for composition. In addition, instead of building a global feature of an unseen class, we use all attribute-based features to form a dense representation consisting of fine-grained attribute details. To recognize unseen classes, we propose a novel training scheme that uses a discriminative model to construct features that are subsequently used to train itself. Therefore, we directly train the discriminative model on composed features without learning separate generative models. We conduct experiments on four popular datasets of DeepFashion, AWA2, CUB, and SUN, showing that our method significantly improves the state of the art.

## 1 Introduction

Zero-shot learning is the important yet challenging task of recognizing unseen class without training samples from samples of seen classes. This setting often arises when dealing with fine-grained recognition problems where some classes have a few or no training samples due to their scarcity [1], such as identifying new fashion trends [2, 3, 4, 5] or endangered species [6, 7, 8, 9, 10, 11]. We argue that classes exhibit compositional structures [12, 13, 14] in which we only need to recognize basic attributes such as color, shape, or material to recognize a large number of classes expressible in terms of these attributes. We propose a zero-shot learning method that reuses attributes of seen classes to construct features of unseen classes for training.

To address the lack of training samples, recent zero-shot works rely upon generative models [15, 16, 17, 18, 19, 20] to synthesize features of unseen classes. These works infer features of unseen classes from features of seen classes. However, methods based on Generative Adversarial Networks [19, 15, 16] suffer from low diversity in generated features. On the other hand, likelihood-based methods [21, 16, 17, 18, 22] promote diversity among generated features, but their generated features are often non-discriminative. These issues are most severe for unseen classes as the feature generation process cannot be regulated without training samples.

Leveraging the remarkable performance of Convolutional Neural Networks [23, 24], most works extract image features by pooling local information from image regions into holistic representations [15, 25, 17, 18, 26, 11]. Although holistic features encode discriminative information among classes,

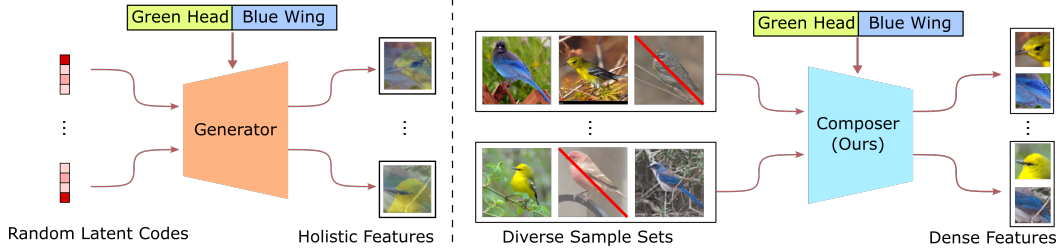

Figure 1: Left: Conventional generative models synthesize holistic features from random codes lacking fine-grained details. Right: Our compositional model constructs dense attribute-based features from training samples. By selecting different relevant samples for composition, our method builds diverse features for unseen classes.

they are not trained to capture attribute details needed for alignment between visual and semantic spaces. Therefore, generating holistic features cannot describe fine-grained details of unseen classes.

Few works have explored learning compositional structures in the few-shot setting [12, 13, 27, 28]. [12, 13] enforce decomposition of representation that requires at least a few samples per class and cannot generalize to unseen classes. Although [28, 29] can compose classifiers for novel concepts, they produce holistic features which fail to preserve attribute details.

**Contributions:** We develop a novel framework that addresses the limitations of aforementioned methods. Instead of generating holistic features as in Figure 1 (left), we extract attribute-based features from seen classes and learn to combine them to effectively construct features of unseen classes, see Figure 1 (right). We augment a discriminative model with a prior distribution to construct features of unseen classes based on class predictions and use these features to update the discriminative model. Our method has several advantages over the state of the art:

– Our framework selectively composes features of unseen classes from semantically related training samples. It also allows specifying different sample sets used for composition that leads to the diversity of composed features. Therefore, we can control the composition process by constraining the samples that are used to build attribute-based features of unseen classes.

– Instead of generating holistic features, which lack fine-grained details, we build a dense feature consisting of attribute-based features that scales to hundreds of attributes.

– Instead of using a generative model to first build features and then train a discriminative model, we use a discriminative model to compose features of unseen classes in order to train itself. This makes the learning process efficient by removing the need for learning additional generative models.

## 2    Related Works

Early works on zero-shot learning focus on learning joint embedding spaces for visual features and class semantics [26, 30, 31, 32, 11]. On the other hand, [33, 34, 35, 36] transfer knowledge from seen to unseen classes via knowledge graphs which encode richer structures than class semantics. However, the predictions of these methods are often biased towards seen classes due to the lack of training samples for unseen classes [37, 38]. To overcome this issue, [39, 40] propose to reduce the probability of seen classes on samples that are out of training distribution, while [41] directly calibrates predictions to favor unseen classes by adding a fixed margin to prediction scores. More recent work [2, 42] propose calibration losses allowing the model to adjust prediction probabilities for unseen classes. However, these works only regularize predictions to prevent bias towards seen classes without effectively transferring knowledge to unseen classes.

To mitigate the differences between training and testing distributions [38, 43], recent zero-shot learning methods [37, 15, 16, 17, 18, 19, 20] use generative models to augment a training set with synthesized features of unseen classes, hence, casting zero-shot learning as a fully supervised learning problem. [37, 44] employ Generative Adversarial Networks (GAN) whose generated features often lack diversity due to mode collapse [45, 46, 47] and lack generalization ability due to memorization of training data [48, 49]. On the other hand, [21, 16, 17, 22] propose to use Variational Autoencoders (VAE) that optimize the likelihood of every training sample to enforce diversity. However, these methods suffer from posterior collapse [50, 51, 52], which results in generating generic non-discriminative features. To address these issues, [16] combines VAE and

GAN, while [18] directly learns a feature generator without any encoder or discriminator. However, these improvements are only effective for seen classes with training samples. Although cycle consistency losses [15, 19, 53] directly regulate the generated features of unseen classes, they often collapse these features together to align them with semantic vectors of their classes, which results in lack of visual diversity among features.

Most works on zero-shot learning [15, 16, 17, 18, 19, 20, 26, 11, 25] rely on holistic image features, which cannot capture local discriminative information from attributes. [54, 7] localize distinct visual parts, which require costly bounding-box annotations. [55, 56, 57, 58] employ attention mechanism to localize discriminative regions in a weakly-supervised setting, however, they cannot capture every attribute due to the limited number of attention models. To overcome this issue, [2] extracts a feature for every attribute by leveraging semantic vectors of attributes. However, it lacks an effective mechanism for transferring attribute knowledge obtained from seen classes to unseen classes.

Decomposing concepts into common components is a natural and simple technique for knowledge sharing [59, 60, 61, 62]. [12, 13] learn compositional representations that generalize to classes with few samples. On the other hand, [27, 63] learn to combine samples for data augmentation. However, these works cannot generalize to unseen classes as they require training samples for every class. While [28, 29, 64, 65] combine attribute classifiers to recognize novel combination of attributes, they build upon holistic features, which cannot capture fine-grained attribute details. Recently, by examining various zero-shot methods, [14] shows that zero-shot generalization depends on the ability to capture attribute information in addition to compositional structure of features.

## 3 Background

We first discuss the problem settings for zero-shot learning. Given that our framework uses attribute-based features as inputs, we then review the recent work in [2], which addresses extracting dense attention features for attributes.

### 3.1 Zero-Shot Learning Problem Setting

We assume that there are two disjoint sets of classes $\mathcal{C}_s$ and $\mathcal{C}_u$, where $\mathcal{C}_s$ denotes seen classes with training images and $\mathcal{C}_u$ denotes unseen classes without any training image. Let $(I_1, y_1), \ldots, (I_N, y_N)$ be $N$ training samples, where $I_i$ denotes the $i$-th training image and $y_i \in \mathcal{C}_s$ corresponds to its class.

Zero-shot learning aims to recognize both seen and unseen classes given only training images of seen classes. In order to generalize to unseen classes, similar to [16, 17, 18], we assume access to *class semantic* vectors of all classes $\{z^c\}_{c \in \mathcal{C}_s \cup \mathcal{C}_u}$ at training time. More specifically, $z^c = [z_1^c, \ldots, z_A^c]^\top$ where $z_a^c$ encodes the strength of attribute $a$ appearing in class $c$. We normalize each $z^c$ to have unit norm to prevent prediction bias toward classes with many attributes. Notice that we also use the semantic vector of each attribute $\{v_a\}_{a=1}^A$ as the average of GloVe representation [66] of each word in attribute names to guide the extraction of attribute-based features [2].

### 3.2 Dense Attention Review

The recent work on Dense-Attention Zero-shot Learning (DAZLE) [2] localizes all attributes in an image and extracts attribute-based features using a *dense attention* mechanism. Let $\{f_i^r\}_{r=1}^R$ be the region features of the image $I_i$ by dividing the image into $R$ equal regions. For the $a$-th attribute, DAZLE computes the *attribute-based feature* $h_i^a$ as the weighted sum of all region features,

$$h_i^a \triangleq \sum_{r=1}^R \alpha(f_i^r, v_a) f_i^r, \quad \alpha(f_i^r, v_a) \triangleq \frac{\exp(v_a^T W_\alpha f_i^r)}{\sum_{r'} \exp(v_a^T W_\alpha f_i^{r'})}, \tag{1}$$

where $W_\alpha$ denotes a learnable matrix measuring the compatibility between attribute semantic vectors and the region features, $\alpha(f_i^r, v_a)$ denotes the importance of a region to an attribute. For an image $I_i$, we define its *dense feature* matrix $H_i \triangleq \begin{bmatrix} h_i^1, & \ldots & , h_i^A \end{bmatrix}$ as the collection of all $A$ attribute-based features. DAZLE uses an *attribute embedding* mechanism to compute the prediction score $s(H_i, z^c)$ as the sum over scores of each attribute,

$$s(H_i, z^c) = \sum_{a=1}^A s(h_i^a, z_a^c), \quad s(h_i^a, z_a^c) \triangleq (v_a^T W_e h_i^a) z_a^c, \tag{2}$$

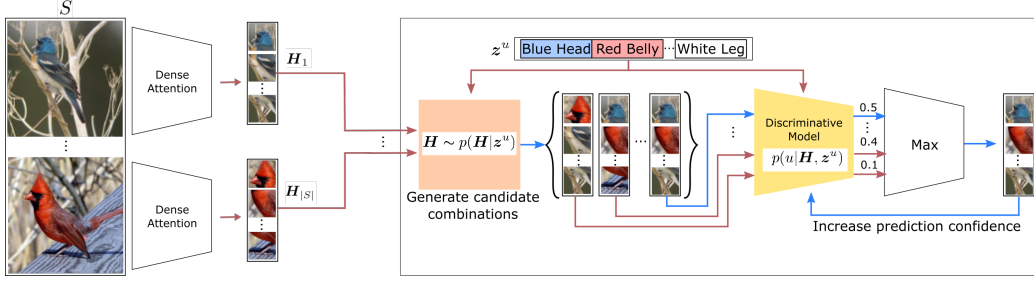

Figure 2: Overview of our compositional zero-shot learning framework. Given a set of samples $S$, we extract dense attribute-based features $\boldsymbol{H}_i$ from each sample $i \in S$. For an unseen class $u$ with a semantic vector $\boldsymbol{z}^u$, we generate candidate combinations by sampling from $p(\boldsymbol{H}|\boldsymbol{z}^u)$ and use a discriminative model $p(y|\boldsymbol{H}, \boldsymbol{z}^u)$ to select the best combination in order to train itself.

where $\boldsymbol{W}_e$ is a learnable matrix capturing the compatibility between attribute-based features and semantic vectors, $s(\boldsymbol{h}_i^a, z_a^c)$ is an attribute score measuring the strength of the attribute $a$ in the image $i$. Finally, the model is trained to minimize the cross-entropy loss between the prediction probability and the ground-truth labels,

$$\min_{\boldsymbol{W}e, \boldsymbol{W}_\alpha, \{\boldsymbol{v}_a\}_{a=1}^A} -\sum_i \log p(y_i|\boldsymbol{H}_i, \boldsymbol{z}^{y_i}), \quad p(y_i|\boldsymbol{H}_i, \boldsymbol{z}^{y_i}) = \frac{\exp(s(\boldsymbol{H}_i, \boldsymbol{z}^{y_i}))}{\sum_{c'} \exp(s(\boldsymbol{H}_i, \boldsymbol{z}^{c'}))}, \quad (3)$$

where $p(y_i|\boldsymbol{H}_i, \boldsymbol{z}^{y_i})$ is the prediction probability calculated by applying softmax normalization on the score $s(\boldsymbol{H}_i, \boldsymbol{z}^{y_i})$. In our framework, we use the dense features from training images $\{\boldsymbol{H}_i\}_{i=1}^N$ as building blocks for our feature composition model. Notice that DAZLE cannot generate features for unseen classes since it only produces classification scores.

The main contribution of our paper is to propose a novel formulation that transforms any discriminative model into a generative model capable of combining attribute-based features from training samples to construct dense features for unseen classes.

## 4 Compositional Zero-Shot Learning via Dense Feature Composition

In this section, we discuss our proposed method for generating dense features, containing all fine-grained features, of unseen classes. Our framework first samples a set of candidate feature combinations from which we select the most probable combination having the largest prediction score $p(u|\boldsymbol{H}, \boldsymbol{z}^u)$ for each unseen class $u$. We develop a unified framework that alternates between constructing features and updating a discriminative model $p(y|\boldsymbol{H}, \boldsymbol{z})$ by increasing its confidence on the features of unseen classes. Algorithm 1 summarizes the steps of our method. We start by defining the compositional property of dense features and propose a framework that generates dense features of unseen classes by leveraging this compositional property.

---

**Algorithm 1** Composing Dense Features

**Input:** Training set $D$, pretrained $p(y|\boldsymbol{H}, \boldsymbol{z})$ on seen classes
**for** $t = 1, \ldots, N_{iteration}$ **do**
 ▷ Construct features for unseen classes
 Sample a set $S \subset D$
 Extract dense features $\{\boldsymbol{H}_i\}_{i \in S}$ via (1)
 Construct $Q_u(S)$ for $u \in \mathcal{C}_u$ via (8)
 Compose $H_u(S)$ from $Q_u(S), \{\boldsymbol{H}_i\}_{i \in S}$ via (10)
 ▷ Update the zero-shot classifier
 Increase $p(u|H_u(S), \boldsymbol{z})$ via (11)
**end for**
**Output:** Optimal $p(y|\boldsymbol{H}, \boldsymbol{z})$

---

### 4.1 Leveraging Compositionality of Dense Features

Our goal is to generate dense features $\boldsymbol{H}$ for a class $y$ with semantic vector $\boldsymbol{z}$. Instead of directly learning and maximizing $p(\boldsymbol{H}|y, \boldsymbol{z})$, which is prone to mode/posterior collapse and cannot scale to a large number of attributes and dense features, our insight is to transform a discriminative model $p(y|\boldsymbol{H}, \boldsymbol{z})$, which is easier to train, into a generative model via application of the Bayes rule,

$$\text{argmax}_{\boldsymbol{H}} \, p(\boldsymbol{H}|y, \boldsymbol{z}) = \text{argmax}_{\boldsymbol{H}} \, p(y|\boldsymbol{H}, \boldsymbol{z}) p(\boldsymbol{H}|\boldsymbol{z}), \quad (4)$$

where the feature prior $p(\boldsymbol{H}|\boldsymbol{z})$ captures the data manifold by assigning high probability to regions containing attribute-based features and small or zero probability otherwise. Due to the high

dimensions of features in $\boldsymbol{H}$, estimating $p(\boldsymbol{H}|\boldsymbol{z})$ is intractable. To overcome this, we propose a compositional assumption on attribute-based features where only features $\boldsymbol{h}_i^a$ from various samples can be combined into a valid $\boldsymbol{H}$. This makes the solution-space more tractable since the number of combinations is finite, yet combinatorial. To enforce this assumption, let $S$ be a subset of training sample indices from which we construct new features (this is the minibatch used during training). We denote by $\mathcal{U}(S)$ the set of all possible combinations of attribute-based features from samples in $S$, i.e.,

$$\mathcal{U}(S) \triangleq \left\{ \left[ \boldsymbol{h}_{i_1}^1, \ \ldots, \ \boldsymbol{h}_{i_A}^A \right] \mid i_a \in S \, \forall a \right\}. \tag{5}$$

We limit the support of the feature prior to only feature combinations in $\mathcal{U}(S)$, i.e.,

$$p(\boldsymbol{H}|\boldsymbol{z}) = 0, \ \boldsymbol{H} \notin \mathcal{U}(S). \tag{6}$$

Selecting features from $\mathcal{U}(S)$ is equivalent to combining attribute-based features across samples in $S$ to describe novel classes, e.g., if S contains samples of a 'blue head, white belly bird' and a 'red bird', $\mathcal{U}(S)$ can describe a variety of birds such as a 'blue head, red belly bird' or a 'red head, white belly bird' (see Figure 2). Thus, we can rewrite (4) as searching for the best feature composition from $\mathcal{U}(S)$ by maximizing

$$\mathrm{argmax}_{\boldsymbol{H}} \, p(\boldsymbol{H}|y, \boldsymbol{z}) \approx \mathrm{argmax}_{\boldsymbol{H} \in \mathcal{U}(S)} \, p(y|\boldsymbol{H}, \boldsymbol{z}) p(\boldsymbol{H}|\boldsymbol{z}). \tag{7}$$

**Remark 1** *Notice that we use class probability predictions as the criteria for selecting feature combinations. Thus, our framework is robust against missing attributes, where some attributes needed to describe a target class are absent from images in $S$, by resorting to the most probable feature combination available in $\mathcal{U}(S)$.*

Next, we discuss an efficient way to solve (7) via sampling. We propose to train the discriminative model directly on the composed features, leading to an efficient framework that does not need to learn a separate generative model.

## 4.2 Composing Dense Features for Unseen Classes

Since optimizing (7) requires an expensive combinatorial search in $\mathcal{U}(S)$, we propose to significantly reduce the search space by avoiding irrelevant combinations. To do so, we impose that $\boldsymbol{H}$ should only be composed from attribute-based features of *semantically related* samples in $S$ with respect to a target unseen class $u$. We define related samples as the ones whose semantic vectors best reconstruct the class semantic vector $\boldsymbol{z}^u$. Let $\mathrm{Q}_u(S)$ be the set set of related samples to unseen class $u$ from $S$,

$$\mathrm{Q}_u(S) \triangleq \mathrm{argmin}_{S' \subseteq S} \left( \min_{\gamma} \| \boldsymbol{z}^u - \sum_{i \in S'} \boldsymbol{z}^i \gamma_i \|_2^2 \right) \quad \mathrm{s.\,t.} \ \ \gamma_i \geq 0, \forall i \in S', \quad |S'| \leq k, \tag{8}$$

where $S'$ denotes any subset of $S$, $\boldsymbol{z}^i$ denotes the ground-truth class semantic vector of the sample $i$, $\gamma_i$ denotes the reconstruction weight, and $k$ is the number of related samples to select. The nonnegative constraint on the combination weights $\gamma_i$'s ensures that the semantic meaning of each sample does not change due to the reconstruction. We solve (8) using Nonnegative Orthogonal Matching Pursuit [67], which greedily adds samples into $\mathrm{Q}_u(S)$ to decrease its loss until no further improvement can be made (see the supplementary materials for more details).

Given the set of related samples $\mathrm{Q}_u(S)$, we construct a prior for attribute-based features such that the more related a sample is to the target semantic vector, the more probable its attribute-based features will be used for composition,

$$p(\boldsymbol{H}|\boldsymbol{z}^u) \triangleq \prod_{a=1}^{A} p(\boldsymbol{h}_{i_a}^a|\boldsymbol{z}^u), \quad p(\boldsymbol{h}_{i_a}^a|\boldsymbol{z}^u) \triangleq \begin{cases} \dfrac{\exp\left(\langle \boldsymbol{z}^{i_a}, \boldsymbol{z}^u \rangle \times T\right)}{\sum_{i \in \mathrm{Q}_u(S)} \exp\left(\langle \boldsymbol{z}^i, \boldsymbol{z}^u \rangle \times T\right)}, & \text{if } i_a \in \mathrm{Q}_u(S), \\ 0, & \text{otherwise,} \end{cases} \tag{9}$$

where $\boldsymbol{z}^{i_a}$ is the ground-truth class semantic vector of sample $i_a$ used to compose attribute $a$, $T$ is a non-negative scalar that controls the probability of using attribute-based features from related samples. When $T \gg 0$, the prior would mostly include attribute-based features from the most related sample and when $T = 0$, it would uniformly sample attribute-based features from all related samples. Notice that we measure the relatedness as the cosine similarity between the sample semantic $\boldsymbol{z}^{i_a}$ and the target semantic $\boldsymbol{z}$. The prior also assigns zero probability to combinations of samples outside the related sample set $\mathrm{Q}_u(S)$ to exclude these combinations. For simplicity, in (9), we assume the

independence of attribute-based features given the semantic vector. However, more general models could be used, which we leave to future work. Indeed, the prior allows us to sample a set of candidate features to find the most probable feature, thus we avoid searching through all combinations in $\mathcal{U}(S)$. Specifically, we first samples a set of candidate combinations $M_u(S)$ from which we seek the combination that maximizes the product of an unseen class probability and the prior,

$$H_u(S) \triangleq \operatorname{argmax}_{\boldsymbol{H} \in M_u(S)} p(u|\boldsymbol{H}, \boldsymbol{z}^u) p(\boldsymbol{H}|\boldsymbol{z}^u), \quad M_u(S) \triangleq \{\boldsymbol{H}|\boldsymbol{H} \sim p(\boldsymbol{H}|\boldsymbol{z}^u)\}, \quad (10)$$

where $H_u(S)$ is the most probable dense feature of class $u$ and $M_u(S)$ is constructed by sampling $b$ candidate combinations according to $p(\boldsymbol{H}|\boldsymbol{z}^u)$[1].

Having the composed features of unseen classes $\{H_u(S)\}_{u \in \mathcal{C}_u}$, we train the discriminative model by increasing its prediction confidence on these features as unseen classes while maintaining the confidence on seen class samples via the following cross-entropy loss,

$$\min_{\boldsymbol{W}_e, \{\boldsymbol{v}_a\}_{a=1}^A} \mathbb{E}_S \left[ -\frac{1}{|S|} \sum_{i \in S} y_i \log p(y_i|\boldsymbol{H}_i \, \boldsymbol{z}^{y_i}) - \frac{1}{|\mathcal{C}_u|} \sum_{u \in \mathcal{C}_u} u \log p(u|H_u(S), \boldsymbol{z}^u) \right]. \quad (11)$$

Here, we transfer knowledge from seen to unseen classes by recognizing novel combinations of features as unseen classes. Thus, the discriminative model learns the existence of unseen classes to avoid seen class bias. We alternate between composing features (10) and minimizing the loss (11) on a random sample set $S$ in each iteration until convergence. By randomizing the sample set $S$, we effectively ensure the diversity among composed features by enforcing them to be built from various sets of samples. Although any classification model can be used as the discriminative model, for simplicity, we reuse DAZLE, pre-trained to extract dense features, as our discriminative model.

For inference, we recognize a test image by finding the most probable class according to the discriminative model: $c^* = \operatorname{argmax}_{c \in \mathcal{C}_s \cup \mathcal{C}_u} p(c|\boldsymbol{H}, \boldsymbol{z}^c)$.

## 5 Experiments

We demonstrate the effectiveness of our framework, referred to as *Composer*, on four popular datasets: DeepFashion [4], AWA2 [68], CUB [69], SUN [70]. We first discuss the datasets, evaluation metrics, implementation details and baselines. We then present the zero-shot and generalized zero-shot performances in pre-trained and fine-tuned feature extractor settings. Finally, we show comparisons between compositional models and current generative models, effects of hyper-parameters, and an ablation study on the challenging DeepFashion dataset.

### 5.1 Experimental Setup

**Datasets:** We conduct experiments on different visual recognition datasets: DeepFashion [4], AWA2 [68], CUB [69], and SUN [70] having different data statistics. DeepFashion [4] contains images of fine-grained clothing categories with 36 seen classes and 10 unseen classes. Due to the redundancy among attributes, we only select top 300 most discriminative attributes by ranking the entropy of each attribute distribution across classes according to [2]. AWA2 [68] is a coarse-grained dataset of animal images with 40 seen classes and 10 unseen classes described by 85 attributes. CUB [69] is a fine-grained bird dataset with 47 images per class for 150 seen classes and 50 unseen classes. Each class is carefully annotated with 312 attributes which can be grounded in images. Finally, SUN [70] is a visual scene dataset with 645 seen classes and 72 unseen classes. However, each class only has 16 images. We follow the data splits of [2] for DeepFashion and of [68] for AWA2, CUB, and SUN.

**Evaluation Metrics:** Similar to [68], we measure the top-1 accuracy on two settings: i) zero-shot learning ($u \to u$) where testing images are from unseen classes thus the model only need to distinguish among unseen classes and ii) generalized zero-shot learning where testing images comes from both seen and unseen classes. In the latter setting, we measure the accuracy when recognizing seen classes ($a \to s$) and unseen classes ($a \to u$). We compute the harmonic mean ($H$) between seen and unseen class accuracy to measure the trade-off between these performances.

| Method | DFashion (5691 images/class) | | | | AWA2 (588 images/class) | | | | CUB (47 images/class) | | | | SUN (16 images/class) | | | |
|---|---|---|---|---|---|---|---|---|---|---|---|---|---|---|---|---|
| | $u \to u$ | $a \to s$ | $a \to u$ | $H$ | $u \to u$ | $a \to s$ | $a \to u$ | $H$ | $u \to u$ | $a \to s$ | $a \to u$ | $H$ | $u \to u$ | $a \to s$ | $a \to u$ | $H$ |
| Pre-trained Setting | | | | | | | | | | | | | | | | |
| MLSE [34] | - | - | - | - | 67.8 | 83.2 | 23.8 | 37.0 | 64.2 | 71.6 | 22.3 | 34.0 | 62.8 | 36.4 | 20.7 | 26.4 |
| CVC [71] | - | - | - | - | 71.1 | 81.4 | 56.4 | 66.7 | 54.4 | 47.6 | 47.4 | 47.5 | 62.6 | 36.3 | 42.8 | 39.3 |
| TripletLoss [72] | - | - | - | - | 67.9 | 83.2 | 48.5 | 61.3 | 63.8 | 52.3 | 55.8 | 53.0 | 63.8 | 30.4 | 47.9 | 36.8 |
| f-VAEGAN-d2 [16] | - | - | - | - | 71.1 | 70.6 | 57.6 | 63.5 | 61.0* | 60.1* | 48.4* | 53.6* | 64.7 | 38.0 | 45.1 | 41.3 |
| CADA-VAE [17] | - | - | - | - | - | 75.0 | 55.8 | 64.0 | - | 53.5 | 51.6 | 52.5 | - | 35.7 | 47.2 | 40.7 |
| f-Translator [18] | 40.7 | 30.5 | 23.9 | 26.8 | 70.4 | 72.6 | 55.3 | 62.6 | 58.5 | 54.8 | 47.0 | 50.6 | 61.5 | 36.8 | 45.3 | 40.6 |
| DAZLE [2] | 38.4 | 38.1 | 21.5 | 27.5 | 67.9 | 75.7 | 60.3 | 67.1 | 65.9 | 59.6 | 56.7 | 58.1 | 60.7 | 24.3 | 52.3 | 33.2 |
| Composer (Ours) | 43.0 | 32.9 | 31.2 | 32.0 | 71.5 | 77.3 | 62.1 | 68.8 | 69.4 | 56.4 | 63.8 | 59.9 | 62.6 | 22.0 | 55.1 | 31.4 |
| Fine-tuned Setting | | | | | | | | | | | | | | | | |
| SMA [54] | - | - | - | - | 68.8 | 87.1 | 37.6 | 52.5 | 71.0 | 71.3 | 36.7 | 48.5 | - | - | - | - |
| LFGAA+SA [73] | - | - | - | - | 68.1 | 90.3 | 50.0 | 64.4 | 67.6 | 79.6 | 43.3 | 64.4 | 61.5 | 34.9 | 20.8 | 26.1 |
| f-VAEGAN-d2 [16] | - | - | - | - | 70.3 | 76.1 | 57.1 | 65.2 | 72.9* | 75.6* | 63.2* | 68.9* | 65.6 | 37.8 | 50.1 | 43.1 |
| AREN+CS [56] | 41.0 | 36.3 | 27.5 | 31.3 | 67.9 | 79.1 | 54.7 | 64.7 | 71.8 | 69.0 | 63.2 | 66.0 | 60.6 | 32.3 | 40.3 | 35.9 |
| DAZLE [2] | 44.1 | 41.3 | 26.5 | 32.3 | 66.7 | 72.1 | 61.7 | 66.5 | 69.7 | 55.4 | 64.1 | 59.4 | 59.5 | 25.0 | 51.5 | 33.7 |
| Composer (Ours) | 47.3 | 42.3 | 32.8 | 36.9 | 75.4 | 76.1 | 62.2 | 68.5 | 74.0 | 61.6 | 66.3 | 63.9 | 61.0 | 24.7 | 53.4 | 33.8 |

Table 1: Performances on DeepFashion, AWA2, CUB and SUN. We report zero-shot accuracy ($u \to u$) in the zero-shot setting and seen class accuracy ($a \to s$), unseen class accuracy ($a \to u$), harmonic mean ($H$) in generalized zero-shot setting. * indicates the usage of extra supervision from human captions.

| Method | AWA2 | | | | CUB | | | |
|---|---|---|---|---|---|---|---|---|
| | $u \to u$ | $a \to s$ | $a \to u$ | $H$ | $u \to u$ | $a \to s$ | $a \to u$ | $H$ |
| Generative models | | | | | | | | |
| f-Translator [18] | 70.4 | 72.6 | 55.3 | 62.6 | 58.5 | 54.8 | 47.0 | 50.6 |
| f-VAEGAN-d2 [16] | 71.1 | 70.6 | 57.6 | 63.5 | 61.0 | 60.1 | 48.4 | 53.6 |
| CADA-VAE [17] | - | 75.0 | 55.8 | 64.0 | - | 53.5 | 51.6 | 52.5 |
| Attribute GANs | 65.1 | 75.2 | 58.1 | 65.6 | - | - | - | - |
| Compositional models | | | | | | | | |
| Random Comp | 65.5 | 76.7 | 56.6 | 65.1 | 67.3 | 64.4 | 51.2 | 57.0 |
| Composer (Ours) | 71.5 | 77.3 | 62.1 | 68.8 | 69.4 | 56.4 | 63.8 | 59.9 |

| Method | DeepFashion | | | |
|---|---|---|---|---|
| | $u \to u$ | $a \to s$ | $a \to u$ | $H$ |
| No Comp | 44.6 | 54.2 | 2.6 | 5.0 |
| Random Comp | 44.9 | 40.4 | 30.1 | 34.5 |
| $p(\boldsymbol{H}|\boldsymbol{z})$ Comp | 43.9 | 36.9 | 37.1 | 36.5 |
| $p(y|\boldsymbol{H},\boldsymbol{z})p(\boldsymbol{H}|\boldsymbol{z})$ Comp (fixed $S$) | 46.9 | 44.7 | 26.9 | 33.6 |
| Composer (Ours) | 47.3 | 42.3 | 32.8 | 36.9 |

Table 2: Left: Comparison between generative models and compositional models on AWA2 and CUB in the pre-trained setting. Right: Ablation study on DeepFashion in the fine-tuned setting.

**Baselines:** We compare our method with 3 main approaches in zero-shot learning: generating features of unseen classes, learning transferable visual representations, and learning compatibility functions. f-VAEGAN-d2 [16] and CADA-VAE [17] generate features of unseen classes using GANs or VAEs while f-Translator [18] directly optimize training data likelihood to learn a feature generator. On the other hand, CVC [71] generates classifiers of unseen classes. To learn visual representation, SMA [54] and AREN+CS [56] use a few attention channels to capture discriminative class features while MLSE [34] learns latent class representations via semantic graphs. DAZLE [2] uses a calibration loss to prevent seen class bias in addition to extract individual attribute-based features. Finally, TripletLoss [72] learns a compatibility function between visual and semantic information by accounting for the semantic similarity among classes. LFGAA+SA [73] proposes a dynamic compatibility function that adapts to attributes appearing in an image. On DeepFashion, we run each baseline using their released codes with their default settings. On the remaining datasets, we use the performances reported in their papers to ensure their best performances.

**Implementation Details:** Following [25], we resize images to $224 \times 224$ and extract features using the ResNet101 backbone [24] for our method. Our setting is comparable with the above baselines except for SMA using VGG19 and LFGAA combining VGG19, GoogleNet, and ResNet101. We use the feature map of the last convolutional layer whose size is $7 \times 7 \times 2048$ and treat it as features from $7 \times 7 = 49$ regions. We implement our framework in PyTorch and optimize it using RMSprop[74] with the default setting, learning rate of $0.0001$ and batch size of $50$ having an equal number of samples per class. We pre-train DAZLE on seen classes and use it to compose dense features for at most 2000 and 4000 iterations, respectively, on a NVIDIA V100 GPU. To prevent seen class bias, we add a margin of 1 to unseen class scores and $-1$ to seen class scores, which reduces the dominance of seen classes similar to [2]. We experiment in two settings: i) using pre-trained ImageNet features (pre-trained setting) and ii) fine-tuning the ResNet backbone on each dataset (fine-tuned setting). To measure the robustness of our method, we fix the hyperparameters at $T = 5, k = 5, b = 50$ ($T = 10, k = 10, b = 50$) for the pretrained (fine-tuned) setting on all datasets.

## 5.2 Experimental Results

**Zero-Shot Learning:** Table 1 shows the zero-shot accuracy ($u \to u$) across different datasets. In the pre-trained setting, our method significantly outperforms other methods by at least 2.3%

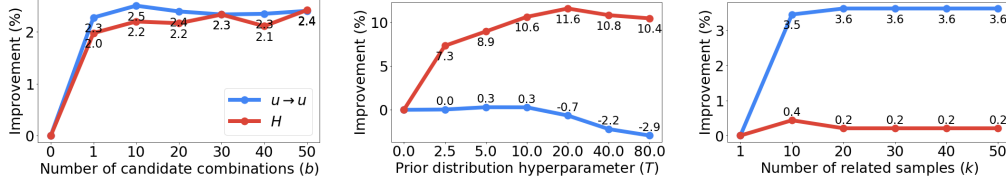

Figure 3: Effects of hyperparameters on harmonic mean ($H$) and zero-shot accuracy ($u \rightarrow u$) on DeepFashion in fine-tuned setting.

and 3.5% on DeepFashion and CUB, respectively while having comparable performance with the state-of-the-art method on AWA2. In the fine-tuned setting, we improve at least 3.2%, 5.1%, and 2.2% on DeepFashion, AWA2, and CUB respectively. Although we do not uses human captions as extra supervision in CUB, our method significantly surpasses `f-VAEGAN-d2` by 8.4% (1.1%) on pre-trained (fine-tuned) settings. Our strong performances demonstrate that dense feature composition can effectively describe fine-grained attribute details of unseen classes. Having only 16 samples per class in SUN does not allow to effectively train dense attention model, which results in low performances.

**Generalized Zero-Shot Learning:** Table 1 also shows generalized zero-shot performances. We observe that methods using dense features, `DAZLE` and ours, surpass other methods on the majority of datasets. Specifically, we improve the harmonic mean by at least 4.5%, 1.7%, 1.8% on DeepFashion, AWA2, CUB, respectively, in pre-trained setting and at least 4.6%, 2.0% on DeepFashion, AWA2, respectively, in fine-tuned setting. Our method achieves high accuracy of seen classes and significantly improves accuracy of unseen classes by 7.3%,1.8%, 6.1% in pre-trained setting on DeepFashion, AWA2, CUB, respectively. Notice that fine-tuning on CUB, SUN with small number of samples overfits to training data due to the high capacity of dense attention.

**Benefits of Dense Feature Composition:** We compare recent generative methods and `Attribute GANs`, where we learn a separate GANs per attribute, with random composition which uniformly samples combinations from $\mathcal{U}(S)$ and our methods in Table 2 (left). Random composition significantly outperforms recent methods by at least 1.1% and 3.4% harmonic mean on AWA2 and CUB, respectively while performing comparably with `Attribute GANs`. We believe this is due to the strong regularization effect of dense feature composition which prevents overfitting on feature combinations from training samples while retaining the ability to recognize fine-grained details obtained from seen classes. Notice that `Attribute GANs` cannot scale to 312 attributes in CUB dataset due to its large memory consumption for training hundreds of GANs, thus we do not report its performance. Our method surpasses random composition by 3.7% (6.0%), 2.9% (2.1%) in harmonic mean (zero-shot accuracy) on AWA2 and CUB, respectively.

**Effect of Hyperparameters:** Figure 3 shows the zero-shot and generalized zero-shot performances on DeepFashion in the fine-tuned setting as functions of $b$, $k$ and $T$. We vary the value of one hyper-parameter while fixing the remaining hyper-parameters and measure the improvement with respect to the lowest value in each hyperparmeter range. By increasing the search budget for most probable combination via the number of candidate combinations $b$, we improve both zero-shot and generalized zero-shot performances compared to random composition ($b = 0$). The performances stabilize across a wide range of $b$ as probable samples according to $p(\boldsymbol{H}|\boldsymbol{z})$ often have high $p(y|\boldsymbol{H}, \boldsymbol{z})p(\boldsymbol{H}|\boldsymbol{z})$ probability. Increasing $T$ increases the similarity between composed features and features of related samples, thus our method constructs "hard" features being closer to the decision boundary of seen classes. The harmonic mean improves with $T$ and degrades for large values of $T$, as the composed features become too similar to training features. When increasing the size of the related samples via $k$, we improve zero-shot accuracy the most, as composed features have richer attribute details by using more samples. Notice that our method only uses at most $k = 20$ related samples, as additional samples are not selected since they do not improve the semantic reconstruction loss (8), thus the performances remain the same for larger $k$.

**Ablation Study:** We report the effectiveness of different components in our method on DeepFashion in Table 2 (right). We observe that the discriminative model, trained only on seen classes, fails to generalize to unseen classes. Although training on random composed features improves the harmonic mean, this does not significantly improve zero-shot accuracy due to the lack of meaningful knowledge

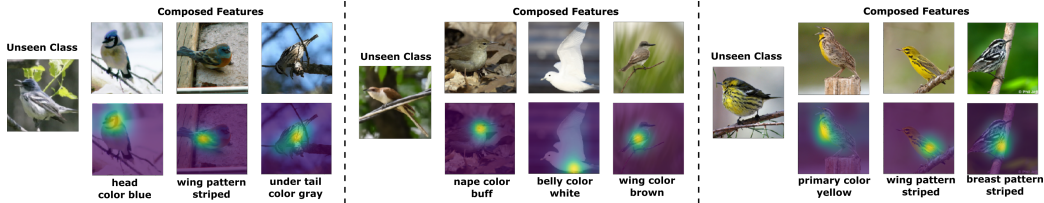

Figure 4: Attention visualization of attribute-based features used for dense feature composition of unseen classes. Our method selects relevant attributes from related samples to describe unseen classes.

in composed features. Using only prior $p(\boldsymbol{H}|\boldsymbol{z})$ improves the harmonic mean by 2.0% but not zero-shot accuracy, as maximizing the prior is equivalent to using features from the most related sample without modifications for unseen classes. We achieve the best performance when combining the classification knowledge $p(y|\boldsymbol{H}, \boldsymbol{z})$ and the prior knowledge $p(\boldsymbol{H}|\boldsymbol{z})$. Notice without varying the sample set $S$, the harmonic mean drops by 3.3%, showing the importance of feature diversity for zero-shot learning.

**Qualitative Results** The generated features of unseen classes can be interpreted by visualizing the attention map of their attribute-based features selected across related samples as shown in Figure 4. We observe that dense attention successfully localizes fine-grained details of attributes in images. Hence, our method selects relevant attributes from related samples to describe unseen classes. Moreover, our method chooses samples with dominant desired attributes for composition such as a white bird for "white belly" or a striped bird for "breast pattern striped".

## 6  Conclusions

We proposed a dense feature composition framework that extracts attribute-based features from training samples and recombines them to construct features of unseen classes. Our framework selectively composes features of unseen classes from only related training samples and alternates between different samples used for composition to improve the diversity among composed features. We employ a novel training scheme where a discriminative model composes features to train itself. By extensive experiments on four popular datasets, we show the effectiveness of our method.

## Broader Impacts

This work addresses the problem of learning without labeled samples, which has fundamental societal, environmental, privacy and technological impacts. Depending less on large-scale annotated data facilitates the process of democratizing machine learning for resource-constrained communities and entities that lack high computational powers or data collection capacity [75]. We also reduce the need for collecting and learning from personal data [76]. Learning without labeled data enables recognition of endangered animal species and plants, and subsequently taking protective measures.

As with any technologies, it is important to study the potential misuses of our method. Since semantic descriptions are often given by humans, methods such as ours could reinforce biases encoded in the semantic information. To prevent biases in predictions, it is important to establish guidelines for regulating and examining the semantic descriptions used for training.

## Acknowledgements

Dat Huynh would like to thank Anh Phong Tran for his valuable comments on the paper first draft. This work is partially supported by DARPA Young Faculty Award (D18AP00050), NSF (IIS-1657197), ONR (N000141812132) and ARO (W911NF1810300).

## Footnotes

[1]Notice that a class can be described by different semantic vectors which reflects its visual variations. For simplicity, we use a single semantic vector per class.

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
