[Supplementary Material]

# Supplementary Materials: Compositional Zero-Shot Learning via Fine-Grained Dense Feature Composition

**Dat Huynh**
Northeastern University
huynh.dat@northeastern.edu

**Ehsan Elhamifar**
Northeastern University
eelhami@ccs.neu.edu

## 1 Finding Related Samples

To evaluate our prior $p(\boldsymbol{H}|\boldsymbol{z})$, we find the set of related samples from $S$ by solving:

$$\operatorname{argmin}_{S' \subseteq S} \mathcal{L}(S') \triangleq \operatorname{argmin}_{S' \subseteq S} \left( \min_{\boldsymbol{\gamma}} \|\boldsymbol{z} - \sum_{i \in S'} \boldsymbol{z}^i \gamma_i\|_2^2 \right) \tag{1}$$

$$\text{s.\,t. } |S'| \le k, \ \gamma_i \ge 0, \ \forall i,$$

where $\mathcal{L}(\cdot)$ is the semantic reconstruction loss for a related sample set $S'$. We employ a greedy nonnegative sparse solver [1] that starts from an empty set of related samples, at each iteration selects the best sample $s$ that minimizes the reconstruction loss $\mathcal{L}$ the most, updates residual errors and repeats to select the next best sample, until $k$ samples are chosen (see Algorithm 1). Let $S'$ denote the set of related samples chosen so far. We compute the optimal reconstruction weights $\boldsymbol{\gamma}^*$ and the residual errors for the target semantic, $\boldsymbol{r}_z$, as

$$\boldsymbol{\gamma}^* = \max\left(0, \arg\min_{\boldsymbol{\gamma}} \|\boldsymbol{z} - \sum_{i \in S'} \boldsymbol{z}^i \gamma_i\|_2^2\right),$$

$$\boldsymbol{r}_z = \boldsymbol{z} - \sum_{i \in S'} \boldsymbol{z}^i \gamma_i^*. \tag{2}$$

For each sample $i$ not in the current set $S'$, we compute the loss $\mathcal{L}(S' \cup \{i\})$ and select the best sample $i$ for which we have the minimum loss value. To do so, we fix the reconstruction weights for samples in $S'$ and compute

$$\mathcal{L}(S' \cup \{i\}) = \min_{\gamma_i} \|\boldsymbol{r}_z - \boldsymbol{z}^i \gamma_i\|_2^2. \tag{3}$$

We select the sample $s$ that achieves the minimum loss value, i.e., $s = \arg\min_i \mathcal{L}(S' \cup \{i\})$. To compute the optimal loss for each $i$, we derive the closed-form of solution of (3) by setting the derivative with respect to $\gamma_i$ to zero,

$$\frac{\partial \mathcal{L}(S' \cup \{i\})}{\partial \gamma_i} = \frac{\partial \|\boldsymbol{r}_z - \boldsymbol{z}^i \gamma_i\|_2^2}{\partial \gamma_i} = 0,$$

$$(\boldsymbol{z}^i)^\top \left( \boldsymbol{z}^i \gamma_i - \boldsymbol{r}_z \right) = 0, \tag{4}$$

$$\implies \gamma_i^* = \frac{\langle \boldsymbol{r}_z, \boldsymbol{z}^i \rangle}{\|\boldsymbol{z}^i\|_2^2}.$$

Substituting (4) into (3), we can compute the optimal loss value for any sample $i$, $\mathcal{L}(S' \cup \{i\})$, as

$$\|\boldsymbol{r}_z - \frac{\langle \boldsymbol{r}_z, \boldsymbol{z}^i \rangle}{\|\boldsymbol{z}^i\|_2^2} \boldsymbol{z}^i\|_2^2 = \frac{\langle \boldsymbol{r}_z, \boldsymbol{z}^i \rangle^2 \|\boldsymbol{z}^i\|_2^2}{\|\boldsymbol{z}^i\|_2^4} - 2 \frac{\langle \boldsymbol{r}_z, \boldsymbol{z}^i \rangle^2}{\|\boldsymbol{z}^i\|_2^2} + \|\boldsymbol{r}_z\|_2^2$$

$$= -\frac{\langle \boldsymbol{r}_z, \boldsymbol{z}^i \rangle^2}{\|\boldsymbol{z}^i\|_2^2} + \text{constant}. \tag{5}$$

Thus, we select the best next sample as

$$s = \arg\max_{i \in S} \frac{\langle \boldsymbol{r}_z, \boldsymbol{z}^i \rangle^2}{\|\boldsymbol{z}^i\|_2^2}. \tag{6}$$

---

**Algorithm 1 : Finding Related Samples via Nonnegative Orthogonal Matching Pursuit**

---

**Input:** $S$: sample set, $\boldsymbol{z}$: target semantic vector, $\{\boldsymbol{z}^i\}_{i \in S}$: semantic vectors of samples, $k$: number of related samples

1: Initialize residuals $\boldsymbol{r}_z = \boldsymbol{z}$, related set $S' = \varnothing$
2: **for** $t = 1, \dots, k$ **do**
3:     $s = \text{argmax}_{i \in S} \frac{\langle \boldsymbol{r}_z, \boldsymbol{z}^i \rangle}{\|\boldsymbol{z}^i\|_2}$
4:     $S' \leftarrow S' \cup \{s\}$
5:     $\boldsymbol{\gamma}^* = \max\big(0, \text{argmin}_\gamma \|\boldsymbol{z} - \sum_{i \in S'} \boldsymbol{z}^i \gamma_i\|_2^2\big)$
6:     $\boldsymbol{r}_z \leftarrow \boldsymbol{z} - \sum_{i \in S'} \boldsymbol{z}^i \gamma_i^*$
7: **end for**
**Output:** related samples set $S'$

---

## 2  Datasets

Table 1 shows the statistics of DeepFashion, AWA2, CUB, and SUN datasets including numbers of attributes and training/testing samples in each dataset.

Table 1: Statistics of the datasets used in our experiments.

| Dataset | # attributes | # seen (val) / unseen classes | # training / testing samples |
|---|---|---|---|
| DeepFashion | 300 | 30 (6) / 10 | **204,885 / 84,337** |
| AWA2 | 85 | 27 (13) / 10 | 23,527 / 13,795 |
| CUB | **312** | 100 (50) / 50 | 7,057 / 4,731 |
| SUN | 102 | **580 (65) / 72** | 10,320 / 4,020 |

## 3  Complexity Analysis

Figure 1 shows memory/space complexity of `f-Translator` which use a generative model to train a discriminative model, and our method which directly trains a discriminative model via self-composition. We measure only training time which excludes loading and evaluation times for 6000 iterations and GPU memory usage. By using a single model for feature generation and classification, we improve the training time by 3 folds and the memory usage by 6 folds compared to `f-Translator`.

Figure 1: Comparison between `f-Translator` and our method in terms of training time and memory usage on DeepFashion dataset.

# 4 Qualitative Results

Figure 2 visualizes attention maps of attribute-based features of present and absent attributes in unseen classes. In addition to using relevant features for present attributes, our method also selects appropriate features to indicate the absence of attributes in unseen classes. Thus, the discriminative model, trained on these features, learns from both present and absent attributes to recognize unseen classes.

Figure 2: Attention visualization of attribute-based features from present and absent attributes in unseen classes.

# References

[1] T. H. Lin and H. T. Kung, "Stable and efficient representation learning with nonnegativity constraints," *International Conference on Machine learning*, 2014. 1