[Reviews · NeurIPS 2020]

Review 1

Summary and Contributions: The authors propose a feature composition framework that learns to extract attribute-based features from training samples and combines them to construct informative features for unseen classes. Instead of building a global feature of an unseen class, the authors use all attribute-based features to form a dense representation that preserves all fine-grained attribute details. Experimental results show that the proposed approach can synthesize diverse samples of unseen classes and improve the classifier's performance.

Strengths: The framework seems reasonable and practical. The features are synthesised by attention method, not the traditional GAN. The authors also make an experimental contribution by providing experiments on several datasets to show the efficacy of their method.

Weaknesses: Major Comments: 1. The proposed model is pretty similar to DAZLE. Authors could highlight the difference and the novelty. 2. Details of some implement are missing, for example, how to classify unseen classes, how to build blocks. 3. As for the images in zero-shot learning datasets, it not always be shot in the perfect angle which may not contain all semantic information (e.g. can get the leg part in the images). In that case, how to sythsis the leg parts for unseen classes? Minor Comments/Typos and expressions: for example,line 144 ("A" should be "a"); line 109 (hard to follow "the strength of having attribute a in class c"? why not "the strength of attribute a in class c")

Correctness: yes

Clarity: Need to be improved. The describtion of the proposed is not very clear.

Relation to Prior Work: need more discussion

Reproducibility: No

Additional Feedback: please refer the weakness above


Review 2

Summary and Contributions: This paper proposed a zero-shot recognition approach by augmenting the training set with synthesized features of unseen classes. Different from previous work, this synthesis is conducted in attributes compositional way without using generative models. Extensive experimental results show that the proposed approach can synthesize more diverse and meaningful features for unseen categories and achieve SOTA performance on several benchmarks.

Strengths: The idea is straightforward and reasonable that exploiting the compositional structure of images. Instead of using the holistic feature, the paper use a set of attribute-level features to compose the feature in a visual-semantic alignment way. To achieve this, the paper proposed the way to directly compose the regional features of seen categories samples. The pipeline is very clean and avoids the unstable caused by using generative models. The proposed self-composing manner further improve the performance. The extensive experimental results show the effectiveness of the proposed approach. The ablation studies show the effectiveness of every component.

Weaknesses: The work is heavily relying on the performance of Dense-Attention Zero-shot Learning to attend the suitable parts of the object. The improvements made by the proposed self-composing way on CUB and DeepFashion are limited. According to the results in Table 1, the performance of the proposed method on seen categories is not so good.

Correctness: Yes, the claims, method, and empirical methodology are correct.

Clarity: Yes, overall the paper is easy to follow.

Relation to Prior Work: Yes, discussions in the related work section are very sufficient. It is clear to see the connections and differences between the proposed work with the previous works that augment the training set with synthesized features of unseen classes, rely on holistic image features, and decompose concepts into common components.

Reproducibility: Yes

Additional Feedback: This work is manipulated in the feature space. The attribute-level features are composed in an arranged agnostic way. In the future, we may consider the space orientation of different attributes and also incorporate diverse background information into the synthesis. ---Post-rebuttal--- I thank author's responses. I think thee compositional way may shed a light into the community. But the current method heavily rely on the "Dense-Attention Zero-shot Learning" which need to be further improved.


Review 3

Summary and Contributions: --- Update after rebuttal --- I thank the author for their detailed rebuttal and effort to clarify the content of the paper and provide missing details. Authors have addressed most pressing concerns, and it is my opinion that their work could be of interest to the community. I would strongly recommend, however, that authors revise the presentation of their manuscript, in particular with respect to clarity/missing details and claims. Please revise/refine the use of certain terms (cf claims about generative models/self-training, see correctness section) and add all the clarifications provided in the rebuttal (in particular with regards to experimental details not provided in the main paper). --- The authors propose a compositional zero shot learning framework than learns to generate training visual features for unseen classes, as a composition of seen image features. The method uses the dense attribute attention method of [10] (DAZLE) to learn a set of attribute specific feature vectors, and subsequently train a classification model by iteratively updating classifier (learning from seen and generated unseen feature) and generating new unseen features using classification predictions. They achieve promising performance, outperforming the state of the art on most datasets, in particular for unseen classification. The main contribution of the proposed work is the concept of generating unseen visual features as a composition of attribute specific seen features. The idea is very interesting and promising but the manuscript needs several clarifications regarding the method, correction of certain claims, and to address missing related work.

Strengths: -The proposed idea of using compositional representations for zero shot learning is very appealing and most likely a requirement for achieving generalisation. The proposed strategy is original. -In particular the idea of using attribute based features as building blocks for the compositional process is smart and intuitive. Similarity the randomness introduced in the sampling process allows to generate a variety of class specific features, affording better training. - Performance is promising especially with respect to the most comparable method DAZLE [10] on which their method in based. The experiments are thorough and the analysis of all model specific parameters is particularly nice.

Weaknesses: -The paper suffers from several weaknesses, in particular with regards to its clarity. The method is described in a very confusing way, making it challenging to understand the building blocks and how they are arranged. Providing e.g. pseudo code describing the whole process, and more specifically the training process would greatly help comprehension. For example: the sampling process for S or where Q(z) computation fits in the whole training process. Figures also lack clarity, in particular Figure 1 which doesn't provide a clear distinction between the 2 compared strategies. - Authors miss an important category of related work relying on class similarity graphs (or knowledge bases) to generate unseen features. Such methods learn to generate features by implicitly exploiting similarities between classes. This contradicts the claim in the introduction l23 stating that there is no existing method leveraging shared concepts across classes. See relation to prior works for more details. - It is mentioned multiple times that the approach uniquely concatenates features instead of averaging them, yet use the classification strategy of DAZLE, which compares attribute specific features to each attribute. Therefore, the feature construction and sampling stage isn't clear. Is a feature vector constructed per attribute? What happens if an attribute is missing? Is S sampled such that all attributes are present?

Correctness: Several claims are confusing or contradicted. Authors mentions multiple time that, in contrast to pre-existing work, they do not learn a generative model yet aim throughout the whole paper to model a joint distribution and to generate image features. Similarly, claims that the classifier is self-trained might require a more subtle description as the model is trained on ground truth labels. The training process is more of the expectation-maximisation type than self training.

Clarity: As mentioned in the weaknesses section, the approach is presented in a confusing way and it is difficult to understand the model construction and training process. The writing is at times overly complicated, using complicated terms to describe a simple process. Simplifying the writing and providing a pseudo-code description of the process would greatly improve comprehension and allow to reproduce the work. More space should have been allocated to Section 4.2, by providing a clearer and more detailed description of how the model is trained and how all components fit together.

Relation to Prior Work: The related work is clearly presented, however an important category of relevant approaches is missing. Authors never mention methods exploiting knowledge bases/graph structures exploiting similarities between classes which have strong performance and generate visual features, similarly to the approaches proposed here. While this work explicitly exploits similarity between classes via the use of attributes, graph based methods generate features by implicitly exploiting class similarities. These approaches need to be discussed and compared to in this work. A particular advantages of these methods over the proposed work is that they do not rely on man-made attributes, which introduce a bias and are not necessarily relevant to identify a category. Below some references related to graph based zero shot learning methods. Zero-shot Recognition via Semantic Embeddings and Knowledge Graphs - Wang et al., CVPR 2018 Multi-Label Zero-Shot Learning with Structured Knowledge Graphs - Lee et al., CVPR 2018 06. Marginalized Latent Semantic Encoder for Zero-Shot Learning - Ding et al., CVPR 2019 Rethinking Knowledge Graph Propagation for Zero-Shot Learning - Kampffmeyer et al., CVPR 2019

Reproducibility: No

Additional Feedback: typos and suggestions: l 19: phrasing suggests that the model doesn't use any training samples, while in practice, seen classes are used to train models. l. 20 typo: endanger -> endangered l. 48. this sentence suggests that the aim of the classifier is to separate real and generated features, not to classify each class individually. l.106. The definition refers to generalised ZSL. l. 131 typo: missing period between classes and Without hyperparameter section: please provide a reminder of what b, T and k correspond to. How does the baseline no Comp relate to the DAZLE model? What is the difference between the 2 approaches? Could the approach been extended to datasets without attributes? (e.g. using learned attributes?)


Review 4

Summary and Contributions: Submission 1107 is concerned with zero-shot learning, the problem that involves recognizing images of classes for which one does not have training samples but class descriptions. The proposed approach heavily relies on the very recent CVPR’20 DAZLE (Dense Attention Zero-Shot Learning) approach. In a nutshell, DAZLE involves (i) computing attribute-based attention features h_i^a (one per image i and per attribute a), then (ii) measuring the compatibility between the h_i^a features and v_a, the GloVe embeddings of the attributes and (iii) computing the final class scores as the weighted sums of the compatibility scores, where the weights depend directly on the attribute/class compatibility (the strength of having attribute a in class c). This work improves over DAZLE by generating attribute-based features h_i^a for unseen classes and training a discriminative model on seen classes (using attribute features of training images) and unseen classes (using generated attribute features). To generate attribute features, a “compositional” assumption is made: the generated features can only be composed by combining real features of semantically related samples. The resulting combinatorial optimization problem is solved using non-negative OMP. Experiments conducted on 4 standard public benchmarks for the zero-shot task (DeepFashion, AWA2, CUB and SUN) show the relevance of the proposed approach.

Strengths: The paper is generally well written. The proposed idea combines the best of attention-based approaches to fine-grained recognition and generation-based approaches to zero-shot learning. The experimental results are quite convincing.

Weaknesses: One of the primary weaknesses of the paper is the assumption that the class semantic vectors z^c are available at training time for both seen *and* unseen classes. While I understand that this assumption has been made by other recent works [15, 16, 17], IMHO this goes against the philosophy of zero-shot learning. Indeed, zero-shot learning makes especially sense for open-ended recognition, i.e. when one has to add new classes on-the-fly and does not have the time to collect images for new classes. Also, if new classes have to be added to the system, then one has to retrain it from scratch, which is particularly inconvenient. Another significant issue is with the comparison with other works that lack details: - First, could the authors please specify how they obtained the results of Table 1 and Table 2 (left) for prior works? By re-running experiments for all of these approaches? Or by copy-pasting results from the respective papers? - Second, could they please clarify which backbone is used in the experiments of these prior works as this may have an important impact on the results? Do all these works use a ResNet 101 backbone as in this work? - Third, the proposed approach as well as DAZLE use additional information wrt other approaches, i.e. the GloVe attribute embeddings. This should be clearly specified in the experiments section.

Correctness: The claims and method appear to be correct.

Clarity: The paper is generally clear and reads easily.

Relation to Prior Work: Good relation to prior work, including many recent references.

Reproducibility: Yes

Additional Feedback: Given that the combinatorial problem is solved with non-negative OMP, does one really need to restrict the set of possible features to semantically-related samples? I would have expected OMP to scale to a large number of samples. Or is the restriction important to improve the accuracy of the approach? Small typo line 185: “to preserves” → “to preserve” Lines 274-275: it is stated that “without using visual information, naïve composition selects samples that may not contain desired attributes (…)”. I though that this would have been the case of the random approach, not the naïve approach (where one composes each attribute-based feature from the sample having the most similar attribute value – cf lines 267-269). Can you please clarify?

[Author Response · NeurIPS 2020]

We would like to thank reviewers for recognizing our original contribution (R7), convincing experiments (R6, 7, 8) and
clarity/reproducibility of the paper (R6, 8). We appreciate suggestions from R6, 7, 8 and will include these in the paper.

**Response to Reviewer 5**

– **C1**: "*lack of novelty: pretty similar to DAZLE*" **Answer:** We respectfully disagree. As mentioned by R7, our
contribution is to generate attribute-based features and compose them for recognition of unseen and seen classes, which
is original and has not been done before, including DAZLE.

– **C2**: "*how to classify unseen classes.*" **Answer:** We use the discriminative model $p(y|\boldsymbol{H}, \boldsymbol{z})$ to compute probabilities
of unseen classes given their semantic vectors $\boldsymbol{z}$ (line 187) and classify a sample as its most probable unseen class.

– **C3**: "*... compared with more state-of-the-art methods.*" **Answer:** As mentioned by R6, 7, 8, our experiments are
extensive. We have included most competitive methods with comparable settings to ours at the submission time.

– **C4**: "*What if the attributes of one unseen class are not shared with seen classes?*" **Answer:** Please notice that ALL
zero-shot learning works rely on sharing attributes between seen and unseen classes. Without sharing common attributes
between seen and unseen classes, there is no way to transfer knowledge to unseen classes for recognition.

**Response to Reviewer 6**

– **C1**: "*The improvements ... on CUB and DeepFashion are limited.*" **Answer:** Please notice that we significantly
improve zero-shot accuracy by at least 3.2% and 2.2% on DeepFashion and CUB, respectively, compared to other
methods (lines 252-253) while having no extra learnable parameter w.r.t. DAZLE.

– **C2**: "*performance of ... seen categories is not so good.*" **Answer:** Please notice that high performance on seen classes
is not the main goal of zero-shot learning as it can cause seen class bias (low unseen class performance). Our method
achieves competitive seen class accuracies while obtaining the best unseen accuracies, hence, high harmonic means.

**Response to Reviewer 7**

– **C1**: "*Providing e.g. pseudo code describing the whole
process ... training process*" **Answer:** Thanks for the
suggestion. We will include the shown Algorithm 1.

– **C2**: "*miss ... work relying on class similarity graphs*"
**Answer:** Thanks. We will discuss these works, if ac-
cepted. Compared to the reported numbers from the most
comparable work (Ding et al., CVPR 2019), our method
outperforms this work in harmonic mean by a significant
margin of 25% on CUB, AWA2 and 5% on SUN.

– **C3**: "*Is a feature vector constructed per attribute? ... Is
S sampled such that all attributes are present?*" **Answer:**
As mentioned in Remark 1, the composed dense features
consist of a feature vector per attribute which is required

---

**Algorithm 1** Self-Composition

1: **Input:** Training set $D$, DAZLE model, ResNet101 backbone
2: Initialize discriminative model $p(y|\boldsymbol{H}, \boldsymbol{z})$ from DAZLE
3: **for** $t = 1, \ldots, N_{iteration}$ **do**
4:     Sample a set $S \subset D$ with an equal number of samples per class (lines 153 and 242)
5:     Extract region features $\{\boldsymbol{f}_i^r\}_{r=0}^R$ for $i \in S$ via ResNet101
6:     Extract dense features $\boldsymbol{H}_i$ for $i \in S$ via DAZLE model
7:     Construct Q$(\boldsymbol{z}^u)$ for $u \in \mathcal{C}_u$ via solving Eq (8) with OMP
8:     Compose features for $u \in \mathcal{C}_u$ based on Q$(\boldsymbol{z}^u)$ via Eq (10)
9:     Update $p(y|\boldsymbol{H}, \boldsymbol{z})$ on real/composed features via Eq (11)
10: **end for**
11: **Output:** Optimal discriminative model $p(y|\boldsymbol{H}, \boldsymbol{z})$

---

for $p(y|\boldsymbol{H}, \boldsymbol{z})$ in DAZLE. Although we try to include samples from various classes having different attributes (line 153,
242), we cannot guarantee $S$ contains all present attributes without extra annotations of present attributes in images.
However, our framework can compose features from any set $S$ by solving Eq (10) even with missing attributes in $S$.

– **C4**: "*How ... no Comp relate to the DAZLE model?*" **Answer:** Please notice that they are different. The No Comp
variant is trained via cross-entropy loss while DAZLE [10] is trained with an extra self-calibration loss (lines 67-68).

– **C5**: "*Could the approach been extended to ... learned attributes?*" **Answer:** For learned attributes, we can factorize
attribute representations into common components (via PCA) which can be used as the building blocks for composition.

**Response to Reviewer 8**

– **C1**: "*semantic vectors $z^c$ are available at training time for both seen \*and\* unseen classes ... [15, 16, 17], ... goes
against the philosophy of zero-shot learning.*" **Answer:** To be comparable with [15 - 17], we follow their setting which
enables our model to compose unseen class features during training, thus the model can learn the testing distributions
with both seen and unseen classes (lines 70-72). Please notice that without the availability of unseen class semantics at
training time, we cannot use self-composition to alternate between training a classifier and composing features.

– **C2**: "*how they obtained the results of Table 1 and Table 2 (left) for prior works?*" **Answer:** Thanks. We will include
the following clarification: On DeepFashion, we run each baseline using their released codes with their default settings.
On the remaining datasets, we use the performances reported in their papers to ensure their best performances.

– **C3**: "*Do all these works use a ResNet 101 backbone*" **Answer:** Except for SMA using VGG19 and LFGAA combining
VGG19, GoogleNet, and ResNet101, all remaining baselines use ResNet101. We will clarify this in the paper.

– **C4**: "*does one really need to restrict the set of possible features to semantically-related samples?*" **Answer:** Please
notice that the related sample set Q$(\boldsymbol{z})$ is required for the sampling probability $p(\hat{\boldsymbol{H}}|\boldsymbol{z})$ and is computed via nonnegative
OMP (lines 171-173 and Algorithm 1). Without related samples, we cannot solve Eq (10) efficiently through sampling.

– **C5**: "*Lines 274-275 ... Can you please clarify?*" **Answer:** Please notice that we set the attribute values of each sample
$\boldsymbol{z}^i$ to its class semantic $\boldsymbol{z}^{y_i}$ (lines 239-240). However, due to occlusion, a sample has many missing attributes compared
to its $\boldsymbol{z}^{y_i}$, thus relying only on $\boldsymbol{z}^{y_i}$ without $\boldsymbol{H}_i$ would compose features lacking many discriminative attributes.

We hope our responses answer the questions and kindly ask the reviewers to raise their scores in light of our responses.

[Meta-Review · NeurIPS 2020]

Initially, this paper received diverging reviews. The reviewers found the idea interesting but had some concerns regarding clarity and the difference between the proposed method and the DAZLE baseline. The authors provided a rebuttal, clarifying the issues that were brought up by the reviews, which satisfied the reviewers. During the discussion, some reviewers have argued that the difference between DAZLE and the paper is clear, and the generated features have been demonstrated to have potential to identify new classes. All reviewers have rated the paper as positive (three "6:marginally above threshold" and one "7:accept") after the discussion phase, so overall the reviewers lean toward accepting. The AC concurs with this decision, and recommends acceptance as a poster. The authors are encouraged to update the paper using the material from the rebuttal to improve the issues brought up during the review, such as clarity.